# Comparison of the Three Most Commonly Used Metabolic Syndrome Definitions in the Chinese Population: A Prospective Study

**DOI:** 10.3390/metabo13010012

**Published:** 2022-12-21

**Authors:** Yilin Huang, Zuo Chen, Xin Wang, Congying Zheng, Lan Shao, Ye Tian, Xue Cao, Yixin Tian, Runlin Gao, Linfeng Zhang, Zengwu Wang

**Affiliations:** 1Division of Prevention and Community Health, National Center for Cardiovascular Disease, National Clinical Research Center of Cardiovascular Diseases, State Key Laboratory of Cardiovascular Disease, Fuwai Hospital, Peking Union Medical College and Chinese Academy of Medical Sciences, Beijing 102308, China; 2Department of Cardiology, National Center for Cardiovascular Disease, National Clinical Research Center of Cardiovascular Disease, State Key Laboratory of Cardiovascular Disease, Fuwai Hospital, Peking Union Medical College and Chinese Academy of Medical Sciences, Beijing 100037, China

**Keywords:** metabolic syndrome, cardiovascular diseases, stroke, China

## Abstract

Metabolic syndrome (MetS) is associated with cardiovascular risk, and there are various definitions, but which is most predictive of future cardiovascular disease (CVD) in the Chinese population is still unclear. MetS was defined with the revised ATP III (Third Adult Treatment Panel Report), International Diabetes Federation (IDF), and the Joint Committee for Developing Chinese Guidelines (JCDCG) definitions. Cox regression was used to estimate the hazard risk of cardiovascular disease among 20,888 participants using the Chinese Hypertension Survey (CHS) data. Sensitivity, specificity, and receiver operating characteristic (ROC) curve distance were used to test the ability of three MetS criteria to identify CVD. During an average follow-up of 4.89 years of 20,888 participants, 925 CVD events occurred (stroke, 560; coronary heart disease, 275; and other cardiovascular events, 119). The revised ATP III criteria identified the most individuals with MetS and had the highest prevalence of MetS. In addition, MetS was associated with a high risk of CVD in both men and women, according to three criteria. The highest diagnostic specificity was for IDF in men and JCDCG in women. The revised ATP III criteria had the highest sensitivity and shortest ROC curve distance in both men and women. Although the MetS definitions, including the revised ATP III, IDF, and JCDCG, are all related to the increased risks of CVD, overall, the revised ATP III performs best and is the most recommended for the Chinese population.

## 1. Introduction

Metabolic syndrome (MetS) is a cluster of inter-related metabolic risk factors, including fat metabolism disorder, obesity, diabetes, insulin resistance, and other risk factors, which help identify individuals at increased risk of type 2 diabetes and cardiovascular disease (CVD) [1]. According to previous studies, MetS doubled the risk of CVD morbidity and mortality worldwide [2,3,4]. A systematic review in China reported that MetS prevalence was 19.2% in men and 27.0% in women, indicating that MetS has become a severe public health problem in China [5].

There are several different definitions of MetS provided by different organizations. In 1998, the World Health Organization first proposed the diagnostic criteria for MetS [6]. The National Cholesterol Education Program Adult Treatment Panel III (NCEP-ATP III) proposed a definition of five components in 2001 [7], the American Heart Association/National Heart, Lung, and Blood Institute (AHA/NHLBI) updated this definition in 2005 [1], the International Diabetes Federation (IDF) recommended a new definition in 2006 [8], and the Joint Committee for Developing Chinese Guidelines (JCDCG) on Prevention and Treatment of Dyslipidemia in Adults suggested a Chinese definition of MetS in 2016 [9]. There were some other definitions from other institutions [10,11]. Existing studies showed that the prevalence of metabolic syndrome varied according to the definitions [12,13,14,15], ranging from 11.5% to 54.2% in China [16,17,18]. In addition, in terms of the association between MetS and cardiovascular disease risk, it also varies by definition [19,20,21]. However, there is no consensus on which diagnosis criteria for MetS is most appropriate for the Chinese population. Some previous studies had recommended the IDF definition of MetS because it was more strongly associated with CHD or acute coronary syndromes than other definitions in the Chinese population, but their validity was limited by using cross-sectional data. In addition, in the Rugao Longevity and Ageing Study in the 70+ age group, the revised ATP III definition was preferred over other definitions due to its higher prevalence and highest association with ischemic stroke or CVD risk [21]. Additionally, existing prospective studies in the Chinese populations were also limited because their study populations consisted exclusively of elderly or occupational subgroups or were conducted only in rural areas or specific communities [19,20,22].

In this paper, our goal was to assess the association of MetS with CVD using data from a prospective study and to compare the predictive value of the most commonly used definitions of MetS for CVD, including the revised ATP III, IDF, and JCDCG, and to identify a more favorable definition for the Chinese and similar populations.

## 2. Results

### 2.1. Baseline Characteristics and Incidence of Cardiovascular Events

Baseline characteristics and incidence of cardiovascular events among study participants are presented in Table 1. The mean age of the study participants was 56.3 years (56.8 years for men and 55.9 years for women). Men had higher rates of smoking, alcohol consumption, waist circumferences, triglycerides, fasting glucose, and systolic and diastolic blood pressure but lower levels of total cholesterol, HDL cholesterol, LDL cholesterol, and BMI. MetS was present in 34.1% of participants (28.2% of men and 39.3% of women) using the revised ATP III criteria, 29.0% (21.9% of men and 35.2% of women) using the IDF criteria, and 22.2% (23.4% of men and 21.1% of women) using the JCDCG criteria. In addition, there was no significant difference between before and after interpolation (Appendix A).

During the median follow-up of 4.89 years, 444 fatal and 481 non-fatal events occurred for a total of 925 CVD events, including 275 CHD events, 560 stroke events, and 119 other cardiovascular events (Table 2).

### 2.2. MetS Defined with Different Definitions and the Risk of CVD

Regardless of which of the three criteria was used, the cumulative risk of CVD was higher in men than in women in both the MetS and non-MetS groups (Figure 1). Effect modification was detected by sex (*p* < 0.001). The trends in cumulative CVD risk curves were almost identical for the three definitions of MetS, differing in that the JCDCG curves for non-MetS men and MetS women overlapped less.

The association of MetS with the risk of cardiovascular events is shown in Table 3. No matter which criteria were used, the incidence of CVD, stroke, and CHD was higher in the population with MetS than without and higher in men than in women. In the total group, any defined metabolic syndrome was significantly associated with CVD, stroke, and CHD with or without model adjustment. Adjusted hazard ratios (HRs) of MetS that were defined according to the revised ATP III, IDF, and the JCDCG criteria for CVD ranged from 1.33 (95% CI 1.11–1.60) (*p* < 0.01) for the revised ATP III to 1.38 (95% CI 1.14–1.67) (*p* < 0.01) for the JCDCG in men and from 1.30 (95% CI 1.01–1.61) (*p* < 0.05) for the JCDCG to 1.37 (95% CI 1.12–1.68) (*p* < 0.001) for IDF in women, respectively. Except for the JCDCG definition in women, MetS defined with all other definitions were significantly associated with the risk of stroke in the adjusted model. Although the association between MetS and CHD defined by almost all definitions was significant in the crude models in both men and women, in the adjusted models, there were significant associations only between MetS defined by IDF (adjusted HR 1.48, 95% CI 1.02–2.17, *p* < 0.05) or JCDCG (adjusted HR 1.59, 95% CI 1.08–2.35 , *p* < 0.05) and CHD in women. Regardless of the definition, elevated blood pressure as well as elevated glucose in the metabolic syndrome components were significantly associated with CVD, stroke, and CHD, as shown in Appendix A.

### 2.3. ROC Analysis

The ROC analysis corresponding to different MetS definitions is shown in Table 4. In the total group, CVD, CHD, and stroke all exhibited similar characteristics, showing the highest sensitivity and negative predictive value as well as the shortest ROC curve distance for the revised ATP III and the highest specificity and positive predictive value for JCDCG. Among three criteria, the sensitivity of the revised ATP III definition of MetS to assess incident CVD, stroke, and CHD was highest at 33.77%, 35.2%, and 36.81% for men and 55.70%, 56.20%, and 58.04% for women, respectively. The specificity of IDF definition of MetS to assess incident CVD, stroke, and CHD for men was highest at 78.31%, 78.30%, and 78.22%, respectively. The specificity of JCDCG definition of MetS to assess incident CVD, stroke, and CHD for women was highest at 79.46%, 79.42%, and 79.20%, respectively. All three definitions had close positive and negative predictive values for CVD, stroke, and CHD, differing by no more than 1%. In the total group, the ROC curve distance of revised ATP III definition to identify CVD, stroke, and CHD was the shortest in total group at 0.6611, 0.6513, and 0.6427, respectively, while it had the highest AUC value (Appendix A).

## 3. Discussion

In the current study, we found that the revised ATP III definition identified the most individuals with MetS and had the highest prevalence of MetS. On the other hand, whether the revised ATP III, IDF, or JCDCG diagnostic criteria were used, MetS was associated with a higher risk of CVD in the Chinese population. Our results also showed that the shortest ROC curve distance in the general population or in men or in women was the revised ATP III definition, whether it was to identify cardiovascular disease, stroke, or CHD.

The revised ATP III criteria identified the most individuals with MetS and had the highest prevalence of MetS, followed by IDF and JCDCG, in agreement with the prior study [23]. Moreover, different diagnostic criteria may show opposite results of the MetS prevalence in different gender. The prevalence of MetS under both the revised ATP III and IDF definitions was higher in women than in men, whereas the prevalence of MetS under the JCDCG criteria was higher in men, which was consistent with previous articles [23,24,25]. This difference between definitions may be due to the high cut-off value of JCDCG for central obesity in women (waist circumference, 80 vs. 85 cm), which could also partly explain why the sensitivity of JCDCG was lowest in identifying cardiovascular disease in women.

A single definition is convenient for the researchers to conduct their research, and it is of specific public health significance to recommend a concept of MetS in clinical practice [26]. Moreover, the greatest significance of MetS is to identify high-risk people for CVD early, and, therefore, its ability to predict risk of disease is key to comparing definitions [27]. Consequently, many existing studies would recommend a definition based on prevalence and its association with CVD. The revised ATP III definition was preferred over other definitions due to its higher prevalence and highest association with ischemic stroke or CVD risk [21,28]. In addition, Wang Q et al. reported that the IDF definition better predicted acute coronary syndrome in Chinese adults [29]. Wang C et al. recommended WHO and JCDCG definitions, with HRs remaining significant when factors of LDL cholesterol and smoking were adjusted [20]. The revised ATP III, IDF, and JCDCG are all applicable to the Chinese population when based only on the degree of association between MetS and CVD. The highest diagnostic specificity was for IDF in men and JCDCG in women, mainly because both are more strictly defined than in the revised ATP III. In both men and women, the revised ATP III-defined MetS performed best in predicting cardiovascular disease, with the highest sensitivity and shortest ROC curve distance. Therefore, the revised ATP III definition had not only the highest prevalence of MetS but also a higher predictive power for CVD in both men and women. On the basis of these considerations, the revised ATP III was preferred.

We found that all definitions of MetS were significantly associated with stroke risk in the adjusted models, except for the JCDCG definition in women, which is consistent with the study by Qian et al. [21]. JCDCG-defined MetS was not significantly associated with stroke in women, possibly due to a higher cutoff value of abnormal glucose (FPG, 6.1 vs. 5.6 mmol/L). Abnormal glucose metabolism showed a high odds ratio (2.73, 95% CI 1.53–4.87) when exploring the association between MetS and stroke [30]. Classifying the high-risk participants into low-risk groups will weaken the association.

There were sex differences in this association, with the strength of the association greater in women than in men, as reported in existing studies [19,31]. However, in our multivariate analysis, the gender differences were significantly reduced and even reversed. To determine the reason for this, we added different covariates into the model separately and found that age was the variable that led to the greatest reduction in the association for women. We further found that the prevalence of MetS was slightly decreasing with age in men, whereas there was a significant increasing trend in women, which may account for the significantly lower association between MetS and CVD in women and slightly higher association in men in the multivariate model. The effect of age on the extent of the association between MetS and CVD in men would become greater as the follow-up period is extended [32].

Our research has several strengths, including the prospective cohort design, large sample size, and rigorous quality control. Second, this is the first national prospective study in China to identify the association as well as to compare definitions between metabolic syndrome and cardiovascular disease risk. Our cardiovascular event outcome endpoints include CVD, CHD, and stroke, providing more comprehensive evidence for the comparison of definitions.

However, there are also some limitations. Firstly, our outcome included both fatal and non-fatal events, and the relationship between MetS and death was not analyzed because of our relatively short follow-up and small number of fatal events. Secondly, MetS status was assessed at baseline, but status of the participants might have changed during the follow-up, which was not considered in this study. Further research is also needed on the relationship between other events and different of definition of MetS.

## 4. Materials and Methods

### 4.1. Design and Study Population

The China Hypertension Survey (CHS) in China was conducted from October 2012 to December 2015, and individuals aged 18 years and older were recruited through stratified multistage random sampling [33,34,35]. First, permanent residents from 262 urban cities and rural counties in all 31 provinces in mainland China were randomly selected. Afterwards, clusters were randomly selected in two stages: regional and rural–urban. Among subjects ≥35 years of age who had completed the baseline survey, 35,000 were randomly selected, of whom 30,036 completed follow-up in 2018–2019 (median follow-up, 4.89 years). The study population for these analyses included subjects who did not meet the following exclusions: pregnant or lactating women (n = 140), prevalent data on CVD at baseline (n = 2131), missing data on MetS components (n = 2425), and those who dropped out in follow-up (n = 5189). After applying these exclusions, 20,888 individuals remained. Figure 2 shows the flow chart of this study.

Written informed consent was obtained from each participant. The Ethics Committee of Fuwai Hospital, Chinese Academy of Medical Science, approved this study.

### 4.2. Data Collection

Data collection was clearly described in the design of CHS [34] Information on demographic characteristics, including age, sex, area, education level, smoking status, alcohol consumption, and family history of cardiovascular disease (CVD) were used. MetS criteria are shown in Table 1. Smoking was defined as participants having smoked at least 20 packs of cigarettes in their lifetime and who currently smoke. Alcohol consumption was defined as having consumed at least one alcoholic beverage per week in the past month. A family history of CVD was defined as self-reported history of hypertension, dyslipidemia, diabetes, coronary heart disease, or stroke in at least one parent and sibling.

Subsequently, anthropometry data (weight, height, and waist circumference) and blood pressure were measured at the local medical centers. Fasting blood samples were collected 10–12 h after the subjects had fasted in the morning, appropriately processed, and immediately refrigerated for analysis at a designated central laboratory (Beijing Adicon Clinical Laboratories, INC, Beijing, China).

### 4.3. Ascertainment of Cardiovascular Outcomes

All cardiovascular events and deaths were followed up in 2018 and 2019 with questionnaires administered in person or by telephone with participants or their proxies. Medical records were further verified for reconfirmation. Following initial documentation by local investigators, hospital records and death certificates were reviewed by the central adjudication committee of Fuwai Hospital (Beijing, China) to determine the final diagnosis. Cardiovascular events were defined as coronary heart disease (CHD, ICD-10 code I20–I25), stroke (I60–I61 and I63–I64), chronic heart failure (I50), and death due to CVD (I00–I25, I27–I88 and I95–I99). Stroke was defined as a diagnosis of subarachnoid hemorrhage, intracerebral hemorrhage, ischemic stroke, or unspecified stroke. CHD was defined as a diagnosis of myocardial infarction, coronary artery bypass graft surgery, or percutaneous coronary intervention.

### 4.4. Statistical Methods

All data analyses were conducted using R version 4.1.3 (http://www.r-project.org (accessed on 12 March 2022). Continuous variables were presented as mean ± standard deviation for those normally distributed and median (lower quartile, upper quartile) for those following a skewed distribution. Categorical variables were presented as frequencies (percentages). Student t-test or Mann–Whitney test and χ2 test were used to assess group differences in continuous and categorical variables, respectively. Incidence rates were stratified by sex and calculated by dividing the number of events by person-time at risk. The cumulative risk of CVD events was assessed using survival curves plots stratified by sex. The proportional risk assumptions were verified using survival curves and Schoenfeld residuals plots. Variables that did not satisfy the proportional risk assumption will be included in the adjusted Cox model as stratification variables.

Multivariable Cox proportional hazards regression was used to model the association between MetS and the risk of CVD. In addition, crude and adjusted models were used. In the crude model, there were no covariates. In the adjusted model, educational level, area, geographical region, smoking, drinking, and family history of CVD were adjusted, and age was included as a stratification variable. The associations of MetS components defined by different criteria with the risk of cardiovascular events are shown in Appendix A. The sensitivity and specificity and the positive and negative predictive values of the incidence of CVD diagnosed by three definitions were calculated. The ROC curve is a two-dimensional curve with sensitivity measures on the Y-axis and specificity on the X-axis (1-specificity) to assess and compare the diagnostic performance of the revised ATP III, IDF, and JCDCG definitions for identifying CVD, stroke, or CHD and is shown in Supplementary Appendix A. The performance of crude model was quantified by the area under the receiver operating characteristic curve (AUC) and compared using DeLong’s test in Appendix A. ROC curve distances were calculated as the distance from the point on the ROC curve to the (0,1) point on the basis of the following equation:$$\mathrm{ROC}\;\mathrm{curve}\;\mathrm{distance}=\sqrt{{(1-\mathrm{sensitivity})}^{2}+(1-{\mathrm{specificity})}^{2}}$$

The shortest distance corresponded to the most suitable MetS diagnostic definitions [36].

Missing values in sex, smoking status, alcohol consumption, education level, BMI, LDL Cholesterol, and Total Cholesterol were addressed using multiple imputations (MI). The comparison of the data set before and after multiple imputations is demonstrated in Appendix A. All tests were two-tailed, and *p* < 0.05 was considered statistically significant.

## 5. Conclusions

The results of the prospective study showed that MetS defined with the revised ATP III, IDF, and JCDCG, were all related to the increased risks of cardiovascular events. However, on the whole, the revised ATP III is the most recommended for the Chinese population.

## Figures and Tables

**Figure 1 metabolites-13-00012-f001:**
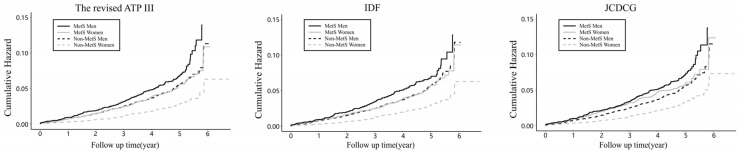
Cumulative hazard of CVD events stratified by sex and MetS. IDF International Diabetes Federation; the revised ATP III, the revised US National Cholesterol Education Program Adult Treatment Panel III; JCDCG, the Joint Committee for Developing Chinese Guidelines; CVD, cardiovascular disease; and CHD, coronary heart disease.

**Figure 2 metabolites-13-00012-f002:**
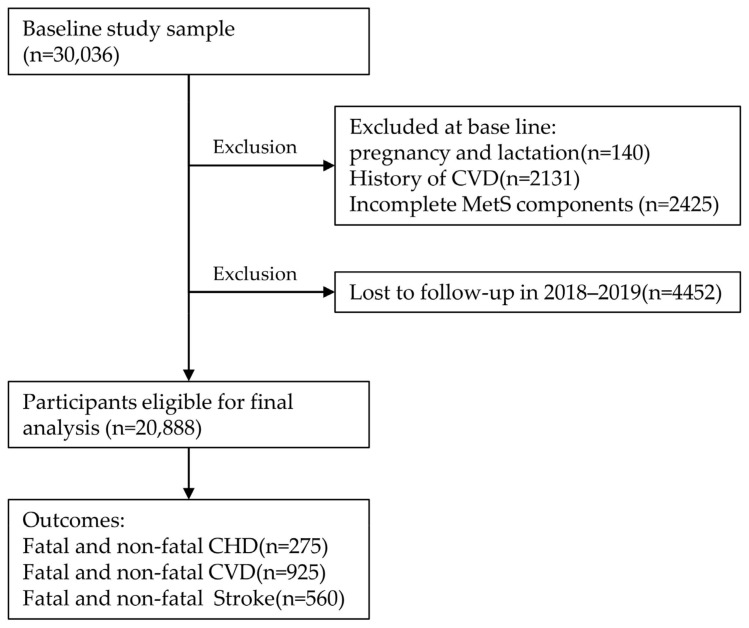
Flowchart of inclusion and exclusion of study participants and cardiovascular outcomes during follow-up. CVD, cardiovascular disease; CHD, coronary heart disease.

**Table 1 metabolites-13-00012-t001:** Criteria for metabolic syndrome.

	AHA/NHLBI, The Revised ATP III	IDF	JCDCG
To be identified as MetS	Any 3 of the following features	Central obesity plus 2 more	Any 3 of the following features
Central obesity	WC ≥ 90 cm for men and WC ≥ 80 cm for women	WC ≥ 90 cm for men and WC ≥ 80 cm for women	WC ≥ 90 cm for men and WC ≥ 85 cm for women
High triglycerides	TG > 1.7 mmol/L or receipt of specific treatment for this lipid abnormality	>1.7 mmol/L or receipt of specific treatment for this lipid abnormality	≥1.7 mmol/L or receipt of specific treatment for this lipid abnormality
Low HDL cholesterol	HDL-C < 40 mg/dL (1.03 mmol/L) in men, HDL-C < 50 mg/dL (1.3 mmol/L) in women, or receipt of drug treatment for reduced HDL-C	HDL-C < 40 mg/dL (1.03 mmol/L) in men, HDL-C < 50 mg/dL (1.29 mmol/L) in women, or specific treatment for this lipid abnormality	HDL-C < 1.0 mmol/l or specific treatment for this lipid abnormality
Elevated blood pressure	BP ≥ 130/85 mmHg or treatment of previously diagnosed hypertension	BP ≥ 130/85 mmHg or treatment of previously diagnosed hypertension	BP ≥ 130/85 mmHg or treatment of previously diagnosed hypertension
Elevated glucose	FPG ≥ 5.6 mmol/dL or drug treatment for elevated glucose	≥5.6 mmol/L or previously diagnosed diabetes mellitus	≥6.1 mmol/L or previously diagnosed diabetes mellitus

IDF, International Diabetes Federation; the revised ATP III, the revised US National Cholesterol Education Program Adult Treatment Panel III; JCDCG, the Joint Committee for Developing Chinese Guidelines; MetS, metabolic syndrome; WC, waist circumference; TG, triglycerides; HDL-C, high-density lipoprotein cholesterol; BP, blood pressure; and FPG, fasting plasma glucose.

**Table 2 metabolites-13-00012-t002:** Baseline characteristics and incidence of cardiovascular events of the study participants.

	Overall (N = 20,888)	Men (N = 9713)	Women (N = 11,175)	*p*-Value
Baseline characteristics				
Age (years)	56.3 ± 13.1	56.8 ± 13.3	55.9 ± 12.9	<0.001
Region (n%)				<0.001
East	8478 (40.6)	3960 (40.8)	4518 (40.4)	
Central	8804 (42.1)	4205 (43.3)	4599 (41.2)	
West	3606 (17.3)	1548 (15.9)	2058 (18.4)	
Area (n%)				<0.001
Urban	9347 (44.7)	4469 (46.0)	4878 (43.7)	
Rural	11541 (55.3)	5244 (54.0)	6297 (56.3)	
Education level (n%)				<0.001
Middle school or below	16650 (79.7)	7388 (76.1)	9262 (82.9)	
High school or vocational school	2922 (14.0)	1567 (16.1)	1355 (12.1)	
College and above	1316 (6.3)	758 (7.8)	558 (5.0)	
Smoking status (n%)	4828 (23.1)	4461 (45.9)	367 (3.3)	<0.001
Alcohol consumption (n%)	4131 (19.8)	3722 (38.3)	409 (3.7)	<0.001
Statin use (n%)	356 (1.7)	158 (1.6)	198 (1.8)	0.450
Waist circumference (cm)	84.1 ± 10.1	85.8 ± 9.94	82.7 ± 10.1	<0.001
Total cholesterol (mmol/L)	4.81 ± 0.971	4.73 ± 0.932	4.88 ± 0.998	<0.001
Triglycerides ^a^ (mmol/L)	1.15 (0.820, 1.700)	1.15 (0.810, 1.77)	1.15 (0.830, 1.66)	0.312
HDL-C (mmol/L)	1.35 ± 0.337	1.31 ± 0.339	1.39 ± 0.331	<0.001
LDL-C (mmol/L)	2.82 ± 0.814	2.76 ± 0.789	2.86 ± 0.832	<0.001
Fasting glucose ^a^ (mmol/L)	5.27 (4.87, 5.79)	5.32 (4.89, 5.85)	5.24 (4.86, 5.73)	<0.001
Systolic BP (mmHg)	133 ± 20.0	133 ± 19.0	132 ± 20.9	<0.001
Diastolic BP (mmHg)	77.8 ± 11.0	79.8 ± 11.1	76.0 ± 10.7	<0.001
BMI (kg/m^2^)	24.6 ± 3.48	24.5 ± 3.38	24.8 ± 3.57	<0.001
Family history of cardiovascular disease (n%)	3130 (15.0)	1288 (13.3)	1842 (16.5)	<0.001
MetS—Revised ATP III defined (n%)	7131 (34.1)	2741 (28.2)	4390 (39.3)	<0.001
MetS—IDF defined (n%)	6058 (29.0)	2125 (21.9)	3933 (35.2)	<0.001
MetS—JCDCG defined (n%)	4633 (22.2)	2275 (23.4)	2358 (21.1)	<0.001
Incidence of cardiovascular events				
Coronary heart disease (n%)	275 (1.3)	163 (1.7)	112 (1.0)	<0.001
Stroke (n%)	560 (2.7)	318 (3.3)	242 (2.2)	<0.001
Cardiovascular disease (n%)	925 (4.4)	530 (5.5)	395 (3.5)	<0.001

In this table, categorical variables are presented by n (%) and were tested by chi-squared tests; normal distributed variables are described as (mean standard deviation) and were tested by *t*-tests. HDL-C, high-density lipoprotein cholesterol; LDL-C, low-density lipoprotein cholesterol; BP, blood pressure; BMI, body mass index; IDF, International Diabetes Federation; Revised ATP III, the revised US National Cholesterol Education Program Adult Treatment Panel III; and JCDCG, the Joint Committee for Developing Chinese Guidelines. a: Skewed distribution variable, described as median (upper quartile, lower quartile), and were tested by Mann–Whitney test.

**Table 3 metabolites-13-00012-t003:** Association of MetS defined using different criteria with the risk of cardiovascular events.

		Cases/PYs (/1000)		HR (95% CI)	
		MetS	Non-MetS	Crude Model	Model 1
Revised ATP III	Cardiovascular disease				
	Total	12.11	8.26	1.47 (1.29, 1.67) ***	1.36 (1.19, 1.56) ***
	Men	14.20	10.91	1.31 (1.09, 1.57) **	1.33 (1.11, 1.60) **
	Women	10.81	5.55	1.94 (1.59, 2.37) ***	1.37 (1.11, 1.67) **
	Stroke				
	Total	7.46	4.87	1.53 (1.30, 1.81) ***	1.44 (1.21, 1.71) ***
	Men	8.78	6.35	1.39 (1.11, 1.75) **	1.46 (1.16, 1.85) **
	Women	6.63	3.35	1.97 (1.53, 2.54) ***	1.37 (1.06, 1.78) *
	Coronary heart disease				
	Total	3.73	2.33	1.60 (1.26, 2.03) ***	1.45 (1.13, 1.85) **
	Men	4.67	3.16	1.48 (1.08, 2.03) *	1.36 (0.98, 1.88)
	Women	3.1	1.48	2.12 (1.46, 3.08) ***	1.43 (0.98, 2.11)
IDF	Cardiovascular disease				
	Total	12.11	8.53	1.42 (1.24, 1.62) ***	1.37 (1.19, 1.58) ***
	Men	14.19	11.18	1.27 (1.05, 1.54) *	1.33 (1.09, 1.62) **
	Women	10.99	5.79	1.90 (1.56, 2.31) ***	1.37 (1.12, 1.68) **
	Stroke				
	Total	7.43	5.06	1.47 (1.24, 1.74) ***	1.43 (1.19, 1.71) ***
	Men	8.68	6.57	1.32 (1.03, 1.69) *	1.43 (1.11, 1.85) **
	Women	6.76	3.49	1.93 (1.50, 2.49) ***	1.39 (1.07, 1.79) *
	Coronary heart disease				
	Total	3.69	2.45	1.50 (1.18, 1.92) ***	1.42 (1.10, 1.82) **
	Men	4.50	3.33	1.35 (0.96, 1.90)	1.26 (0.89, 1.79)
	Women	3.25	1.53	1.93 (1.50, 2.49) ***	1.48 (1.02, 2.17) *
JCDCG	Cardiovascular disease				
	Total	13.40	8.49	1.59 (1.38, 1.83) ***	1.36 (1.18, 1.57) ***
	Men	14.81	10.94	1.36 (1.13, 1.64) **	1.38 (1.14, 1.67) **
	Women	12.05	6.45	1.88 (1.52, 2.32) ***	1.30 (1.05, 1.61) *
	Stroke				
	Total	7.93	5.13	1.55 (1.30, 1.86) ***	1.34 (1.12, 1.61) **
	Men	9.12	6.40	1.43 (1.13, 1.82) **	1.50 (1.18, 1.92) **
	Women	6.79	4.07	1.68 (1.28, 2.20) ***	1.14 (0.86, 1.50)
	Coronary heart disease				
	Total	4.31	2.39	1.81 (1.41, 2.32) ***	1.49 (1.15, 1.92) **
	Men	4.72	3.24	1.45 (1.04, 2.03) *	1.32 (0.94, 1.85)
	Women	3.92	1.66	2.35 (1.61, 3.45) ***	1.59 (1.08, 2.35) *

* *p* < 0.05, ** *p* < 0.01, *** *p* < 0.001; HR, hazard ratio; CI, confidence interval; Pys, person-years; IDF, International Diabetes Federation; the revised ATP III, the revised US National Cholesterol Education Program Adult Treatment Panel III; and JCDCG, the Joint Committee for Developing Chinese Guidelines. Adjusted Model: adjusted for educational level, area, geographical region, smoking, alcohol consumption, and family history of CVD, and whether age > 65 as stratifying variable. Additional adjustment for sex in Total group.

**Table 4 metabolites-13-00012-t004:** The ROC curve analysis for the revised ATP, IDF, and JCDCG definitions as diagnostic tests for CVD, stroke, and CHD.

	Total			Men			Women		
	The Revised ATP III	IDF	JCDCG	The Revised ATP III	IDF	JCDCG	The Revised ATP III	IDF	JCDCG
Cardiovascular disease									
Sensitivity	43.14%	36.65%	30.70%	33.77%	26.23%	29.06%	55.70%	50.63%	32.91%
Specificity	66.28%	71.35%	78.35%	72.10%	78.37%	77.04%	61.32%	65.37%	79.46%
Positive predictive value	5.60%	5.60%	6.17%	6.53%	6.54%	6.81%	5.01%	5.09%	5.55%
Negative predictive value	96.18%	96.05%	96.06%	94.97%	94.85%	94.95%	97.42%	97.31%	97.00%
ROC curve distance	0.6611	0.6953	0.7260	0.7186	0.7688	0.7456	0.5881	0.603	0.7016
Stroke									
Sensitivity	44.29%	37.50%	30.36%	35.22%	27.04%	30.19%	56.20%	51.24%	30.58%
Specificity	66.28%	71.23%	78.18%	72.02%	78.30%	76.95%	61.09%	65.16%	79.24%
Positive predictive value	3.48%	3.47%	3.69%	4.09%	4.05%	4.24%	3.10%	3.15%	3.16%
Negative predictive value	97.73%	97.64%	97.60%	97.05%	96.94%	97.02%	98.44%	98.37%	98.10%
ROC curve distance	0.6513	0.6880	0.7298	0.7057	0.7612	0.7352	0.5859	0.5993	0.7246
Coronary heart disease									
Sensitivity	45.45%	38.18%	33.82%	36.81%	27.61%	30.67%	58.04%	53.57%	38.39%
Specificity	66.01%	71.12%	78.11%	71.93%	78.22%	76.84%	60.91%	64.99%	79.20%
Positive predictive value	1.75%	1.73%	2.02%	2.19%	2.12%	2.21%	1.48%	1.53%	1.83%
Negative predictive value	98.91%	98.85%	98.88%	98.52%	98.44%	98.48%	99.31%	99.28%	99.22%
ROC curve distance	0.6427	0.6823	0.6971	0.6915	0.7560	0.7309	0.5735	0.5815	0.6502

The revised ATP III, the revised US National Cholesterol Education Program Adult Treatment Panel III; IDF, International Diabetes Federation; and JCDCG, the Joint Committee for Developing Chinese Guidelines.

## Data Availability

The dataset analyzed during the current study is available from the corresponding author on reasonable request. The data are not publicly available due to privacy concerns.

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
