# Peer review of "Comparison of the Three Most Commonly Used Metabolic Syndrome Definitions in the Chinese Population: A Prospective Study"

_metabolites, 2022, doi:10.3390/metabo13010012_

Round 1

Reviewer 1 Report

Current definitions of metabolic syndrome (Mets) vary. To clarify the most appropriate definition of Mets in the Chinese population, Huang et al used the Chinese Hypertension Survey (CHS) data to address this issue. Overall, the manuscript and its data were of high quality. ROC and Cox regression confirmed the best applicability of the revised ATP III in the Chinese population at different endpoints for different genders. Overall, I think the manuscript is of sufficient content and quality to warrant publication in metabolites.

Author Response

Thank you very much for your recognition of our work!

Reviewer 2 Report

This is a prospective study to compare the predictive value of different definitions of Metabolic Syndrome (MetS) for cardiovascular disease (CVD).

The authors compared 3 diagnostic criteria (Line 49 includes typo): ATP III, IDF, and JDCGG to determine the best one for Chinese population.

Since China has the largest population in the world, the sampling must have been challenging. In Line 63 authors wrote: "individuals aged 18 and older were recruited through stratified multistage random sampling". It is not clear if the study had the age limit to be recruited. It is also not clear if the healthy adults were targeted, if the individuals were blindly recruited, or if the volunteered individual were involved in this study.  These may cause bias.

In Line 64 the authors wrote: "permanent residents randomly selected from 262 urban cities and rural counties were enrolled across all 31 provinces in mainland China." It would have been difficult to recruit participants evenly from all the provinces in such a big country. How did they even out ?

Please clarify that there is no (or minimal) selection bias in this study.

In Line 106 the authors wrote: "CHD was defined as a diagnosis of myocardial infarction, coronary artery bypass graft surgery, percutaneous coronary intervention." How about angina and arrhythmia? Vasospastic angina and VT/VF may also cause death.

Reviewer 3 Report

Huang and colleagues used data from the Chinese Hypertension Survey (CHS) to investigate the impact of metabolic syndrome (MetS) on cardiovascular risk in a sample of 20,888 people. Their analysis assessed the presence of MetS using the three most commonly used definitions (ATPIII, IDF, and JCDCG). Based on their results, MetS identified by all three definitions had a higher rate of cardiovascular events than individuals without MetS.  They also found that ATP III performed best and suggested that this should be applied to the Chinese population.

Comments/suggestions:

1.       The title is inaccurate, I suggest a change.

2.       The Introduction does not sufficiently explain the reasons behind the research. I recommend revising it.

3.       For Cox regression, in addition to the MetS, it may be worth looking at the association of the MetS components with the output.

4.       Several factors were adjusted for in the analyses, but not for gender in the combined (male and female together) group. In any case, adjustment for sex is also necessary, as cardiovascular risk differs between them.

5.       In Table 1, which describes the general characteristics, the comparison is between male and female sex. However, the main focus of the manuscript is the association of the development of cardiovascular disease and MetS. A comparison of groups with and without a cardiovascular event would be more relevant.

6.       The manuscript includes several supplements. These should be referred to in the main text.

7.       In Table 2, the different p-value thresholds would be presented in a more uniform way: *<0.05, **<0.01, and ***<0.001.

8.       The final conclusions of the manuscript are closely related to the results of the ROC analysis presented in Table 3. Missing from this table is the joint analysis of the two sexes. Also, the results are not considered robust. For neither group did the sensitivity and specificity exceed 80%, and the area under the ROC curves presented in the supplementary material also appears to be low (the value of which is not shown in the figure or in the manuscript).

9.       The AUC values are statistically comparable. It would be worthwhile to perform such an analysis to demonstrate whether the ATPIII definition used to determine MetS is indeed and significantly stronger predictor than the other two.

Overall, the results presented in the manuscript do not differ from those presented in previous publications. It is well known that metabolic syndrome is associated with higher cardiovascular risk. The results of the ROC curve analysis are inconclusive, and I suggest that more complex analyses (including MetS components) should be performed.

Round 2

Reviewer 3 Report

I accept the authors' answers to my questions/comments.